# Fantastic Frogs and Where to Use Them: Unveiling the Hidden Cinobufagin’s Promise in Combating Lung Cancer Development and Progression Through a Systematic Review of Preclinical Evidence

**DOI:** 10.3390/cancers16223758

**Published:** 2024-11-07

**Authors:** Sandra Maria Barbalho, Karina Torres Pomini, Enzo Pereira de Lima, Jéssica da Silva Camarinha Oliveira, Beatriz Leme Boaro, Adriano Cressoni Araújo, Elen Landgraf Guiguer, Rose Eli Grassi Rici, Durvanei Augusto Maria, Jesselina Francisco dos Santos Haber, Virgínia Maria Cavallari Strozze Catharin, Patrícia Cincotto dos Santos Bueno, Eliana de Souza Bastos Mazuqueli Pereira, Ricardo de Alvares Goulart, Lucas Fornari Laurindo

**Affiliations:** 1Department of Biochemistry and Pharmacology, School of Medicine, Universidade de Marília (UNIMAR), Marília 17525-902, SP, Brazil; smbarbalho@gmail.com (S.M.B.); karinatorrespomini@unimar.br (K.T.P.); enzopereiradelima2020@gmail.com (E.P.d.L.); araujo01@unimar.br (A.C.A.); eleng@unimar.br (E.L.G.); patriciabueno@unimar.br (P.C.d.S.B.); ricardogoulartmed@hotmail.com (R.d.A.G.); 2Postgraduate Program in Structural and Functional Interactions in Rehabilitation, School of Medicine, Universidade de Marília (UNIMAR), Marília 17525-902, SP, Brazil; roseeli@usp.br (R.E.G.R.); elianabastos@unimar.br (E.d.S.B.M.P.); 3Department of Biochemistry and Nutrition, School of Food and Technology of Marília (FATEC), Marília 17500-000, SP, Brazil; 4UNIMAR Charity Hospital, Universidade de Marília (UNIMAR), Marília 17525-902, SP, Brazil; 5Department of Biochemistry and Pharmacology, School of Medicine, Faculdade de Medicina de Marília (FAMEMA), Marília 17519-030, SP, Brazilbeatrizboaro@hotmail.com (B.L.B.); 6Graduate Program in Anatomy of Domestic and Wild Animals, College of Veterinary Medicine and Animal Science, University of São Paulo, São Paulo 05508-220, SP, Brazil; 7Development and Innovation Laboratory, Butantan Institute, São Paulo 05585-000, SP, Brazil; durvanei.maria@butantan.gov.br; 8Department of Odontology, School of Odontology, Universidade de Marília (UNIMAR), Marília 17525-902, SP, Brazil; 9Department of Administration, Associate Degree in Hospital Management, Universidade de Marília (UNIMAR), Marília 17525-902, SP, Brazil

**Keywords:** cinobufagin, non-small-cell lung cancer (NSCLC), preclinical models, apoptosis, molecular pathways, bufadienolide, anticancer therapy, tumor inhibition

## Abstract

Green healthcare relates to using naturally derived sources as medicines to treat and personalize treatments for various diseases. Cancer is one primary global health concern due to its rapid evolution and high prevalence, especially lung cancer. Cinobufagin (CB), a bufadienolide derived primarily from the parotid glands of frogs, has shown promise in combating lung cancer. Our objective with this systematic review is to synthesize the current evidence on CB’s effects against lung cancer, focusing on its mechanisms of action, efficacy, and potential clinical implications. Our results indicated that CB reduces lung cancer tumor growth via increased apoptosis by reducing cancer cells’ viability. In addition, CB also has impacted migration and invasion across multiple lung cancer cell lines and xenograft models. The molecular pathways involved Bcl-2, Bax, cleaved caspases, caveolin-1, FLOT2, Akt, STAT3, and FOXO1. CB achieved these effects with minimal toxicity.

## 1. Introduction

Cancer remains one of the most critical global health challenges [1,2], characterized by the uncontrolled proliferation of cells that invade surrounding tissues [3] and metastasize to distant organs [4]. This uncontrolled cell growth results from a complex interplay of genetic mutations [5] and epigenetic alterations [6] that disrupt normal cellular functions [7], including those governing cell division [8], differentiation [9], and apoptosis [10]. These disruptions facilitate the formation of malignant tumors, which not only compromise the integrity of their origin tissues [11] but also possess the potential to spread throughout the body [12], making treatment increasingly difficult [13].

The treatment landscape for cancer is complicated by the disease’s inherent diversity [14] and adaptability [15]. Despite significant advancements in therapeutic approaches [16], challenges such as drug resistance [17], severe side effects [18], and variable efficacy across different cancer types persist [19]. Consequently, there is a pressing need for novel therapeutic agents that specifically target cancer cells, enhance the efficacy of current treatments, and reduce adverse effects [20].

Cinobufagin (CB) (Figure 1), a bufadienolide found in traditional Chinese medicine [21] and derived from the parotid glands of toads [22], has recently garnered attention for its potential anticancer properties [23,24]. Bufadienolides are a class of steroidal compounds with documented biological activities [25], including anti-inflammatory [26], antiviral [27], and anticancer effects [28]. CB’s potential as an anticancer agent is supported by its ability to induce apoptosis [29], inhibit cell proliferation [30], and modulate vital molecular pathways involved in cancer progression [31]. CB derivatives also possess anticancer effectiveness against lung cancer cells, especially the carbamate derivatives [32] (Figure 1). Figure 2 illustrates the main anticancer effects of bufadienolides.

CB’s ability to induce apoptosis in cancer cells is particularly noteworthy, as apoptosis is a critical mechanism for eliminating malignant cells and preventing tumor growth [34]. Additionally, CB influences cell cycle regulation, which can lead to cell cycle arrest [35] and inhibit the uncontrolled proliferation characteristic of cancer [36]. It also affects cancer cell migration [37] and invasion [24], indicating potential in reducing metastasis [38]. Furthermore, CB interacts with several crucial molecular pathways implicated in cancer. It modulates pathways such as LEF1 Lymphoid-enhancer-binding Factor 1 (LEF1) [39], Mitogen-activated Protein Kinase (MAPK) [40], Signal Transducer and Activator of Transcription 3 (STAT3) [41], Aurora Kinase A-Rapamycin Kinase–Eukaryotic Translation Initiation Factor 4E (AURKA-mTOR-eIF4E) [42], and Phosphoinositide 3-Kinase/Protein Kinase B (PI3K/Akt) [43]. These pathways play significant roles in regulating cell survival, growth, and metabolism, and CB’s impact on these pathways suggests a multifaceted approach to cancer treatment.

This manuscript provides a comprehensive review of CB’s anticancer effects, specifically in lung cancer, focusing on preclinical evidence due to the current absence of clinical studies. To our knowledge, it represents the first detailed analysis exclusively dedicated to CB’s impact across various types of lung cancer. Our review is critical because it offers an in-depth examination of CB’s effects using various preclinical models, enhancing the analysis’s breadth and depth. We assess CB’s influence on critical aspects such as lung cancer cell viability, apoptosis, migration, and molecular pathway modulation. This thorough examination aims to elucidate CB’s mechanisms of action and its potential as a therapeutic agent in lung cancer. In addition to presenting the current preclinical evidence, our manuscript will discuss the possible clinical implications of CB’s anticancer effects. We will also outline future research directions to advance understanding of CB’s role in cancer therapy. Moreover, we address the limitations of the included studies, providing a balanced perspective on the current evidence and identifying areas where further investigation is needed. Through this comprehensive approach, we seek to offer valuable insights that could guide future research and potentially inform clinical applications of CB in lung cancer therapy.

## 2. Literature Search Methodology

To investigate the efficacy and mechanisms of CB in preclinical models, we conducted a comprehensive literature search using several databases, including PubMed, Scopus, Web of Science, and Google Scholar. The search strategy incorporated keywords such as “CB”, “preclinical models”, “experimental studies”, “cancer”, “lung cancer”, and “metastasis”, alongside terms related to specific biological outcomes and processes like “apoptosis”, “cell proliferation”, “molecular signaling”, and “therapeutic efficacy”. The inclusion criteria were specifically tailored to focus on experimental studies involving CB and utilizing preclinical models of lung cancer. Studies were needed to present data on CB’s impact on lung cancer outcomes, relevant molecular mechanisms, and therapeutic potential. To ensure the relevance of the research, only studies that used experimental designs, reported transparent methodologies and results, and involved preclinical models of lung cancer were considered. Exclusion criteria included non-experimental papers, reviews, meta-analyses, and studies unrelated to CB or preclinical models. Studies published in languages other than English or those that did not meet the rigorous standards of experimental research were excluded. Two researchers (S.M.B. and L.F.L.) conducted data extraction using a standardized form to ensure consistency. Essential information collected included study design, details of CB administration (such as dosage and duration), outcomes measured, and any limitations reported. The quality of each study was assessed based on experimental design, sample size, and the clarity of result reporting, adhering to established guidelines for scientific quality. The data were synthesized qualitatively to summarize the effects of CB across various preclinical models, identify common findings, and discuss any limitations. This approach also sought to highlight gaps in the current knowledge and propose directions for further studies.

## 3. Literature Search Report

Records were sourced from multiple avenues to ensure comprehensive coverage of the topic. Specifically, 207 records were initially identified from various electronic databases, and an additional 56 records were obtained from research registers. Before screening, several records were removed to refine the dataset. A total of 48 duplicate records were eliminated to avoid redundancy. Furthermore, automation tools marked 89 records as ineligible based on predefined criteria, and 59 records were removed for other reasons, such as evident irrelevance to the research topic. After removing duplicates and ineligible records, 67 were screened based on their titles and abstracts. Of these, 57 records were excluded for not meeting the inclusion criteria. Ten reports were then sought for full-text retrieval to determine their suitability for inclusion in this review. All ten reports were successfully retrieved and assessed for eligibility. During this assessment, three reports were excluded: one was a non-experimental paper, one did not involve CB, and one was not based on a preclinical model. Ultimately, seven studies met all the inclusion criteria and were included in the review. This process ensured that the selected studies were relevant, experimental, focused on CB, utilized preclinical models, and were published in English, thus maintaining the rigor and quality of the review. Figure 3 provides a visual summary of the literature search process, illustrating the flow of records through the different stages of identification, screening, eligibility assessment, and inclusion.

## 4. CB: Unveiling the Hidden Bufadienolide’s Promise in Combating Lung Cancer Development and Progression

Table 1 presents a comprehensive overview of various studies investigating the effects of CB on lung cancer models. The data include in vitro and in vivo experiments across multiple lung cancer cell lines and animal models, highlighting CB’s potential as a therapeutic agent. The findings showcase CB’s ability to reduce cell viability, induce apoptosis, and inhibit tumor growth and metastasis through various mechanisms, such as targeting specific apoptotic pathways and lipid raft components. Despite promising results, the limitations of each study—such as reliance on particular cell lines, limited in vivo validation, and challenges in translating results to clinical settings—underscore the need for further research. This comparative analysis aims to provide insights into CB’s therapeutic potential and guide future investigations toward effective and safe treatment strategies against lung cancer.

Xu et al. [45] explored the impact of CB on Non-Small-cell Lung Cancer (NSCLC) cell lines (A549, H1299) and the non-cancerous 16HBE cells. The study found that CB significantly reduced cell viability and induced apoptosis in NSCLC cells through mechanisms involving decreased B-cell lymphoma 2 (Bcl-2) levels and increased apoptotic markers such as Bcl-2-associated X Protein (Bax) and cleaved caspases. Additionally, CB affected lipid raft components by reducing caveolin-1 and Flotillin-2 (FLOT2) expression and inhibiting Protein Kinase B (Akt) phosphorylation. However, the study’s reliance on in vitro models and a limited number of cell lines might not entirely reflect in vivo tumor dynamics. Future research should address these limitations by incorporating more diverse NSCLC cell lines and utilizing advanced animal models. Furthermore, integrating immunological therapies, such as immune checkpoint inhibitors, with CB could enhance its therapeutic efficacy. Genetic studies could also explore the impact of CB on specific genetic mutations associated with NSCLC to tailor personalized treatment strategies.

Yan et al. [46] investigated CB’s effects in both in vitro NSCLC models (PC-9, H460) and an in vivo mouse xenograft model. The study demonstrated that CB effectively reduced cell viability, proliferation, and migration, inducing apoptosis and inhibiting tumor growth. However, the research is limited by its use of a single xenograft model and lack of long-term safety data. In order to build on these findings, future studies should explore the effects of CB in combination with existing chemotherapeutic agents and assess its potential in synergistic therapies. Additionally, incorporating radiotherapy alongside CB could provide a multi-modal approach to enhance overall treatment outcomes. Expanding research to include clinical trials will be essential to confirm CB’s efficacy and safety in diverse patient populations.

Sun et al. [47] focused on the H1299 NSCLC cell line, revealing that CB effectively inhibited cell proliferation, migration, and invasion with a low IC_50_ value. The study also identified CB’s impact on molecular pathways such as integrin α2, β-catenin, Focal Adhesion Kinase (FAK), and Proto-Oncogene Tyrosine-protein Kinase Src (Src), which are crucial for NSCLC metastasis. While the in vitro results are compelling, the study lacks in vivo validation and long-term safety data. Future research should address these gaps by incorporating diverse in vivo models and evaluating CB’s effects on tumor microenvironments. Additionally, exploring CB’s role in combination with targeted therapies and immunotherapies could provide a comprehensive approach to tackling NSCLC. Investigating genetic variations and their interaction with CB might offer insights into personalized treatment options.

Zhang et al. [48] evaluated CB’s effects on the A549 NSCLC cell line, the BEAS-2B standard cell line, and a mouse xenograft model. The study found that CB suppressed A549 cell proliferation in a dose- and time-dependent manner with no cytotoxic effect on BEAS-2B cells. CB promoted apoptosis in A549 cells, significantly inhibited migration and invasion, and modulated molecular pathways by increasing Forkhead box O1 (FOXO1) and decreasing Euchromatic Histone Lysine Methyltransferase 2 (G9a) expression. In vivo, CB inhibited A549 xenograft tumor growth, migration, and invasion while promoting apoptosis. However, the study was limited to A549 cells and a single mouse model, and BEAS-2B cells were used for cytotoxicity comparison but not for mechanistic insights. The mechanisms involving FOXO1 and G9a may not be universally applicable across all NSCLC subtypes. Future research should further explore CB’s potential as a targeted cancer therapy by examining its effects on other NSCLC subtypes and investigating the broader applicability of the FOXO1 and G9a pathways.

Li et al. [49] examined CB-loaded polydopamine (PDA) nanomedicine, which enhanced the solubility and targeting of CB in lung cancer cells. The study showed that this nanomedicine significantly inhibited tumor growth in xenograft models compared to controls. However, the clinical applicability remains uncertain due to the study’s focus on in vitro and animal models and potential scalability issues. Future research should prioritize scaling up the nanomedicine for clinical trials, focusing on its safety and efficacy in human subjects. Additionally, combining this nanomedicine approach with other therapeutic modalities, such as targeted gene therapies and immunological agents, could enhance its effectiveness. Investigating the potential for CB-loaded nanomedicines to improve the delivery and efficacy of existing chemotherapies and radiotherapies should also be a priority.

Peng et al. [50] conducted a comprehensive study on CB, evaluating its effects on various lung cancer cell lines and normal bronchial epithelial cells and its in vivo antitumor efficacy. Their findings revealed that CB exhibits substantial anticancer activity. CB effectively inhibited cell growth, migration, and invasion and induced apoptosis through caspase activation and mitochondrial fragmentation. The study also highlighted that CB significantly increased Reactive Oxygen Species (ROS) production in cancer cells, with ROS scavengers reducing its apoptotic effects, underscoring the role of oxidative stress in CB’s mechanism of action. In vivo, CB demonstrated strong antitumor efficacy by inhibiting tumor growth and activating Tumor Protein p53 (p53) phosphorylation in a nude mouse xenograft model. Despite these promising results, the study’s limitations include using a single animal model and focusing on specific cell lines, which may not fully represent the heterogeneity of NSCLC. Future research should aim to validate these findings across different NSCLC subtypes and incorporate long-term safety assessments. Additionally, exploring CB’s potential with other therapeutic modalities, such as targeted therapies and immunotherapies, could provide a more comprehensive treatment strategy for lung cancer.

Zhang et al. [51] evaluated CB’s effects on multiple NSCLC cell lines (A549, H1299, H460, SK-MES-1) and in NSG mice. The study found that CB effectively induced cell cycle arrest, increased apoptosis, and significantly inhibited tumor growth. However, the study’s limitations include using a single mouse model and a restricted range of cell lines, which may not fully represent NSCLC heterogeneity. Future research should include a broader array of preclinical models to capture the heterogeneity of NSCLC and evaluate CB in combination with existing treatments such as chemotherapy and targeted therapies. Investigating the potential of CB to enhance responses to immunotherapies and radiotherapies could provide a more comprehensive treatment strategy. Genetic studies could also explore the impact of CB on various genetic mutations and pathways involved in NSCLC, leading to more personalized treatment approaches.

In summary, the findings across these studies highlight CB’s potential as a novel treatment for lung cancer through its effects on apoptosis, signaling pathways, and tumor growth inhibition. To translate these results into effective clinical therapies, future research should address the limitations of current studies by incorporating diverse preclinical models, investigating synergistic therapies, and conducting comprehensive clinical trials. Emphasizing immunological therapies, genetic studies, and the integration of CB with existing and emerging treatment modalities will be crucial in advancing CB’s therapeutic potential and improving outcomes for lung cancer patients.

CB’s toxicity has garnered significant attention and concern regarding its safety as a potential drug candidate. However, little evidence is investigating this compound’s toxicological and pharmacokinetic characteristics. Wei et al. reported CB’s pharmacokinetics in rats by designing a method for rapidly and accurately determining CB’s blood concentration after intragastric administration of 20 mg/kg in rats. The maximum concentration was 45.83 ± 4.56 ng/mL, the time for maximum concentration was 0.083 ± 0 h, and the half-life was 2.79 ± 0.93 h. Although these authors did not evaluate the toxicological aspects of CB against normal, healthy cells, these results indicate that CB was rapidly absorbed and eliminated [52,53]. In a separate study, Baek et al. demonstrated that treatment with 0.5 and 1 μM of CB for 24 h promoted higher cytotoxicity against human multiple myeloma cells than in peripheral blood mononuclear cells, inhibiting cancer cell viability but not normal cells [40]. Similar results were found in Yin et al.’s study [54]. These researchers reported that CB exerted cytotoxicity against cultured osteosarcoma cells but less or no toxicity to human osteoblast cell lineage. These cells were treated with CB solutions for 12, 24, and 48 h.

Unlike other chemotherapeutic agents that demonstrate cytotoxicity against normal, healthy cells, CB does not induce oxidative Deoxyribonucleic Acid (DNA) damage and cytotoxicity against healthy cells at sublethal doses. Despite conventional DNA-damaging chemotherapies that target all cells, CB explores a cancer vulnerability, a common feature of most malignant transformed cells, which may be a metabolic vulnerability. It is well documented that cancer cells present additional sensitivity to oxidative insult due to high intrinsic oxidative pressure and diminished spare antioxidant capacity. However, cancerous lineages may survive toxic attacks by upregulating DNA repair to maintain genome functions, and CB-induced oxidative damage leads to replication stress and DNA damage response activation [55,56,57]. Combining or conjugating nontoxic doses of CB with chemotherapeutic agents that target DNA response to genotoxic attacks may be a potential avenue for future research endeavors for enhancing CB’s anticancer efficacy and reducing the potential side effects of synthetic chemotherapy drugs since lower doses of synthetic medications may be necessary due to the synergism with CB. Another point that must be raised is that CB can trigger anaphylactoid reactions by binding to the immunoglobulin E (IgE) receptor (IgE-R) and releasing β-hexosaminidase and histamine by this trigger [58]. Research has also shown that CB can interfere with the immune system, ultimately modulating the inflammatory phenotype of immune cells [59].

## 5. Conclusions and Future Research Directions

This manuscript has explored CB’s promising therapeutic potential in lung cancer treatment (Figure 4). Through various studies, CB has been shown to exert significant anticancer effects by inducing apoptosis, inhibiting tumor growth, and modulating vital molecular pathways in lung cancer progression. Despite these promising results, several critical areas warrant further investigation to fully realize CB’s potential as a treatment for lung cancer.

Exploring CB’s impact on the tumor microenvironment (TME) is also a crucial direction for future research. Current studies have demonstrated CB’s direct cytotoxic effects on lung cancer cells, yet understanding how CB interacts with the TME could offer new insights into its overall therapeutic efficacy. The TME, which includes various cell types, extracellular matrix components, and soluble factors, significantly influences tumor progression and therapeutic resistance. Future research should investigate how CB affects immune cell infiltration, cytokine profiles, and interactions between tumor cells and stromal components. Such studies could pave the way for integrating CB with immunotherapies or other treatments to reshape the TME to enhance therapeutic outcomes.

Additionally, evaluating the long-term effects of CB and the potential development of resistance is crucial. Initial findings highlight CB’s effectiveness, but understanding how prolonged exposure might lead to resistance is essential. Research should focus on identifying molecular pathways involved in resistance and exploring genetic and epigenetic changes in lung cancer cells. This knowledge could guide the development of strategies to overcome resistance, potentially involving combination therapies or alternating regimens to maintain CB’s efficacy over time.

Another critical area for future research is assessing CB’s role in metastatic lung cancer. While current studies indicate CB’s effectiveness in inhibiting primary tumor growth, its effects on metastasis have not been thoroughly examined. Future investigations should explore how CB influences the tumor cell migration, invasion, and colonization of distant organs. Evaluating CB’s impact on metastatic pathways and circulating tumor cells could provide valuable insights into its potential as a treatment for metastatic disease. Combining CB with therapies targeting metastatic progression could further enhance its efficacy and improve patient outcomes.

Exploring the synergistic effects of CB with radiotherapy presents a promising research avenue. Radiotherapy is a standard treatment for lung cancer, and integrating CB could offer complementary benefits. Future studies should explore how CB interacts with radiation, including its impact on radiation-induced DNA damage repair and tumor cell apoptosis. Identifying optimal dosing schedules and treatment combinations could help maximize CB’s and radiotherapy’s therapeutic benefits, leading to more effective treatment regimens.

Developing advanced delivery systems for CB could also significantly enhance its therapeutic potential [60]. While initial studies have explored CB-loaded nanomedicines, future research should focus on refining these delivery methods to improve targeting accuracy and minimize off-target effects. Innovative delivery systems, such as ligand-targeted nanoparticles or encapsulation techniques, could enhance CB’s bioavailability and distribution within tumors. In targeted therapies, antibody–drug conjugate (ADC) strategies have been named the “biological missile” to target cancerous lineages effectively. ADC’s strategy is to combine a monoclonal antibody with a cytotoxic drug by a chemical linker. This results in advantages for precise targeting ability and potent killing effect, achieving an accurate and efficient elimination of cancerous cells [61]. Combining CB with monoclonal antibodies to target cancer lineages may be a potential avenue for future research endeavors since ADC is the leading targeted cancer therapy. Additionally, investigating patient-specific factors affecting drug delivery could lead to more personalized treatment approaches, optimizing CB’s benefits for individual patients.

Moreover, incorporating principles of green health into future research on CB can align therapeutic advancements with environmental sustainability. Green health emphasizes the development of eco-friendly and sustainable technologies, which is increasingly important in modern drug development. Research into environmentally responsible manufacturing processes for CB and its delivery systems could reduce the ecological footprint of cancer therapies. For instance, utilizing biodegradable materials in drug delivery systems or developing less toxic synthesis routes for CB could contribute to greener healthcare practices. This approach not only promotes the development of effective treatments but also supports broader goals of environmental stewardship and public health [62,63,64,65].

Finally, mitigating the potential toxicity of CB to leverage its therapeutic benefits for lung cancer treatment is entirely warranted. While CB demonstrates significant anticancer activity, its safety profile remains a critical concern. Investigating the mechanisms underlying CB’s toxicity and developing strategies to minimize adverse effects is essential. This could involve identifying biomarkers for early detection of toxicity, optimizing dosing regimens, and employing drug delivery systems that target tumors more precisely while reducing systemic exposure. Additionally, exploring combination therapies that enhance CB’s efficacy while mitigating its side effects could improve its safety and therapeutic index. Addressing these toxicity concerns through rigorous preclinical and clinical studies will be crucial for advancing CB from promising research findings to a viable and safe treatment option for lung cancer patients.

Exploring these research directions will be crucial in advancing CB as a therapeutic agent for lung cancer. By addressing these areas, we can potentially enhance treatment efficacy, overcome resistance, and improve outcomes for patients with lung cancer. Additionally, integrating green health principles into research and development efforts will ensure that advancements in cancer therapy are aligned with sustainable and environmentally friendly practices, promoting a holistic approach to cancer treatment and ecological responsibility. Concerning mechanisms of action, CB targeting lung cancer appears to involve the upstream or downstream of molecular pathways associated with cancer cell division, death, and metabolism, although these pathways are not seen to be modulated in all models concurrently. Table 2 demonstrates the genetic targets of CB to facilitate the presumptive assumptions of mechanisms of action’s interactions.

## Figures and Tables

**Figure 1 cancers-16-03758-f001:**
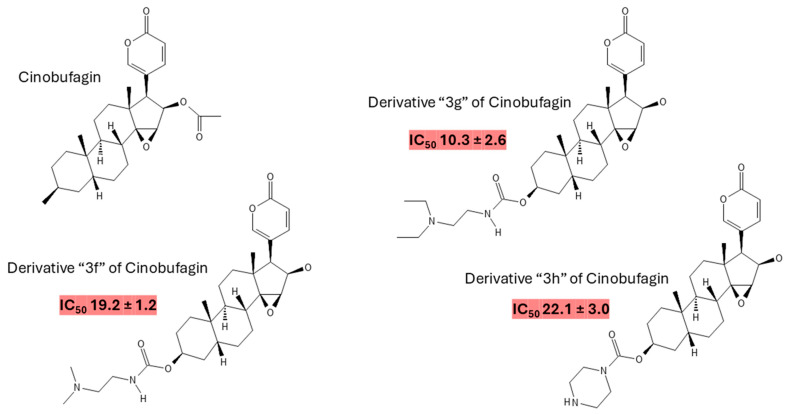
Molecular structure of CB. CB is a bufadienolide with a molecular weight of 442.5 g/mol and an exact mass of 442.23553880 g/mol. The monoisotopic mass is the same, reflecting its precise atomic composition. Its lipophilicity is indicated by an XLogP3-AA value of 3.3, suggesting moderate solubility in lipids and fats. The compound has one hydrogen bond donor and six hydrogen bond acceptors. It features three rotatable bonds, which can contribute to its conformational flexibility. CB’s topological polar surface area (TPSA) is 85.4 Å^2^, which helps understand its interaction with polar environments. Additionally, CB has a heavy atom count of 32 and a complexity score of 923, reflecting its intricate structure. There are ten defined stereocenters, crucial for their three-dimensional orientation, while no undefined stereocenters or defined/undefined bond stereocenters exist. The compound is canonicalized, ensuring its chemical structure is represented in a standard format [33]. The figure also represents carbamate derivatives of CB with anticancer activity against A549 lung cancer cells alongside their respective IC_50_ values (nM) for 72 h [32].

**Figure 2 cancers-16-03758-f002:**
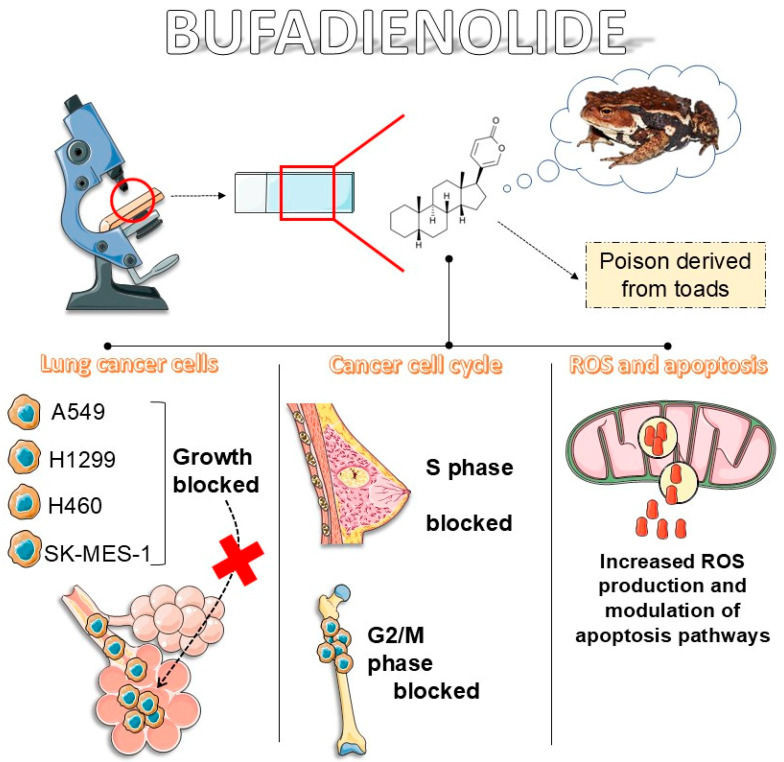
Main anticancer properties of bufadienolides.

**Figure 3 cancers-16-03758-f003:**
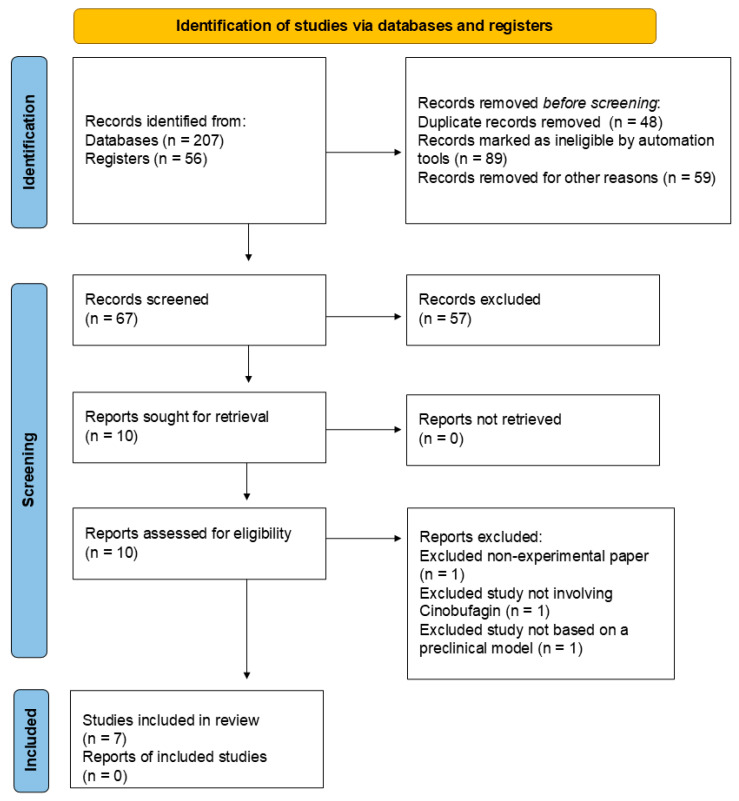
PRISMA flow diagram of the literature search process [44].

**Figure 4 cancers-16-03758-f004:**
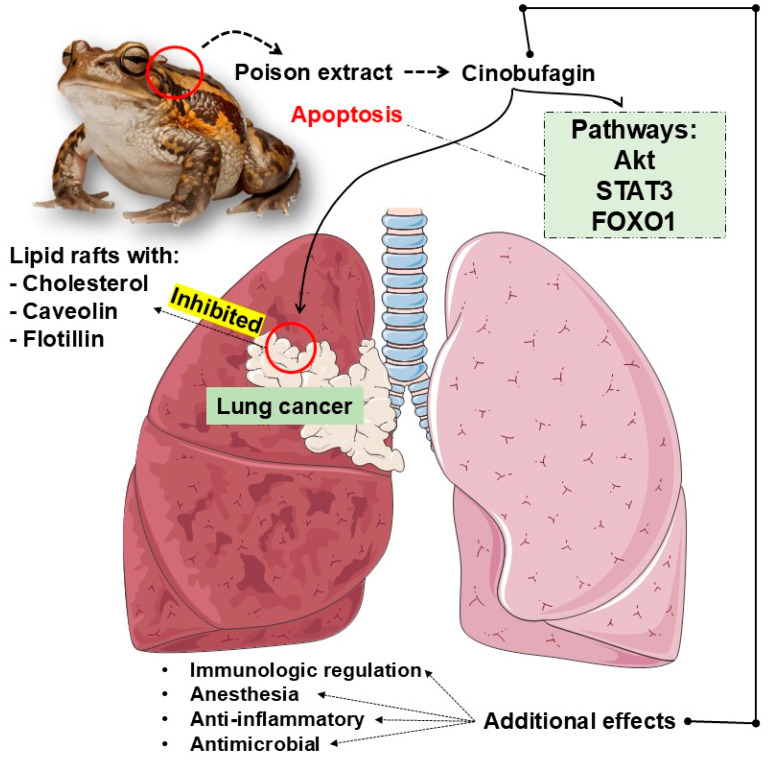
CB’s promising therapeutic potential in lung cancer treatment. Akt, Protein Kinase B; FOXO1, Forkhead box O1; STAT3, Signal Transducer and Activator of Transcription 3.

**Table 1 cancers-16-03758-t001:** Exploring the efficacy and mechanisms of CB in lung cancer intervention and treatment: a comparative analysis of in vitro and in vivo studies.

Model	Intervention	IC_50_	Studies’ Outcomes	Studies’ Limitations	Possible Clinical Implications	Ref.
NSCLC cells (A549, H1299) and 16HBE cells	CB: 0.5, 1, and 2 µM	Not reported	CB significantly reduced viability in NSCLC cells; no significant effect on 16HBE cells. Increased apoptosis in NSCLC cells, dose-dependently; Bcl-2 levels decreased, Bax, cleaved caspase-3, and cleaved caspase-9 levels increased. Increased ROS levels in NSCLC cells. Redistributed cholesterol and sphingomyelin; reduced caveolin-1 and FLOT2 expression; inhibited p-Akt.	Results are based on in vitro models, which may not fully reflect in vivo tumor dynamics and immune interactions. Limited to two cell lines, which may not represent all NSCLC types.	CB could offer a novel treatment for NSCLC by targeting lipid rafts and caveolin-1, potentially overcoming drug resistance and enhancing therapy effectiveness. Clinical trials are needed for validation.	[45]
NSCLC cells (PC-9, H460) and mouse xenograft model	CB: 0.1, 0.2, and 0.5 µM for in vitro studies; 0.5–2 mg/kg/day via intraperitoneal injection for in vivo studies	0.5624 μM and 0.04657 μM for PC-9 and H460 (NSCLC cells)	**In vitro:** CB significantly reduced cell viability, proliferation, and colony formation; induced apoptosis; inhibited migration and EMT. **In vivo:** CB significantly inhibited tumor growth in xenograft models; reduced phosphorylation of STAT3 and reversed IL-6-induced STAT3 nuclear translocation. No significant toxicity was observed.	Reliance on in vitro and a single xenograft model; the study does not address long-term safety or possible clinical applicability in humans.	CB effectively targets STAT3 signaling, showing promise as a therapeutic agent for NSCLC. Further clinical trials are necessary to confirm its efficacy and safety in human patients, and to explore potential in combination therapies.	[46]
H1299 NSCLC cell line	CB: 0.01, 0.04, and 0.16 µM	0.035–0.008 μM for H1299 cells	CB inhibited H1299 cell proliferation. Significant antiproliferative activity was also observed in colony formation assays. CB significantly reduced DNA synthesis. CB significantly inhibited migration and invasion of H1299 cells. Concentration-dependent reduction in the expression of integrin α2, β-catenin, FAK, Src, c-Myc, and STAT3 was observed.	Limited to a single NSCLC cell line may not fully represent the heterogeneity of NSCLC. The study primarily focuses on in vitro assays, lacking in vivo validation. The long-term effects and safety profile of CB are not addressed.	CB shows potential as an effective agent against both the growth and metastasis of NSCLC by targeting critical molecular pathways involved in cell proliferation and migration. This suggests CB could be a valuable candidate for further development in NSCLC treatment strategies.	[47]
A549 NSCLC cell line, BEAS-2B normal cell line, mouse xenograft model	CB: 0–120 nM for in vitro studies; 10 mg/kg/day via intraperitoneal injection for in vivo studies	25.15 ± 0.88 nM (24 h), 18.29 ± 1.28 μM (48 h), and 10.46 ± 1.79 nM (72 h) for A549 cells and 176.04 ± 31.50 nM (72 h) for BEAS-2B cells	**In vitro:** CB suppressed A549 cell proliferation in a dose- and time-dependent manner, with no cytotoxic effect on BEAS-2B cells. CB promoted apoptosis in A549 cells. CB significantly inhibited migration and invasion in A549 cells. Increased FOXO1 expression and decreased G9a expression in A549 cells. **In vivo:** CB inhibited A549 xenograft tumor growth, migration, and invasion while promoting apoptosis.	The study was limited to A549 cells and a single mouse model; BEAS-2B cells were used for cytotoxicity comparison but not for mechanistic insights. Mechanisms involving FOXO1 and G9a may not be universally applicable across all NSCLC subtypes.	CB could be a promising candidate for targeted cancer therapy in NSCLC by modulating FOXO1 and G9a. The findings highlight CB’s potential to inhibit NSCLC progression and suggest further investigation for its therapeutic application.	[48]
BEAS-2B, A549, and LLC cell lines, mouse xenograft model	CB-loaded PDA-based nanomedicine: 0–80 nM for in vitro studies; 1–2 mg/kg/day via intraperitoneal injection for in vivo studies	61 nM for free CB and 32 nM for CB-loaded PDA-based nanomedicine in A549 cells; 74 nM for free CB and 39 nM for CB-loaded PDA-based nanomedicine in LLC cells	**In vitro:** The PDA nanomedicine demonstrated improved solubility and bioavailability of CB. Enhanced targeting of lung cancer cells via folic acid-receptor interaction was observed. **In vivo:** In a xenograft model, CB-loaded PDA nanomedicine inhibited tumor growth significantly compared to controls.	The study was conducted primarily in vitro and in animal models; the clinical translation of the nanomedicine’s effectiveness and safety needs further validation. Potential scalability issues and complex synthesis might affect practical application.	This targeted and responsive nanomedicine could enhance the effectiveness of CB in lung cancer therapy by improving its bioavailability and reducing side effects. The approach offers a promising strategy for delivering chemotherapeutic agents with precision and controlled release, potentially advancing treatment options for lung cancer. Further research and clinical trials must confirm its efficacy and safety in human subjects.	[49]
NSCLC cells (A549, NCI-H460, H1299, SK-MES-1, Calu-3) and BEAS-2B cells; nude mice xenograft model	CB: 0.5–4.0 μM for in vitro studies; 2.5–5.0 mg/kg/day via intraperitoneal injection for in vivo studies	Values ranging from 2.3 to 6.7 μM for A549, NCI-H460, H1299, SK-MES-1, and Calu-3; 22.3 μM for BEAS-2B	**In vitro:** CB significantly inhibited growth, migration, and invasion of A549 cells. CB-induced apoptosis through caspase activation and mitochondrial fragmentation. **In vivo:** CB demonstrated strong antitumor efficacy by inhibiting tumor growth and activating p53 phosphorylation.	The study results are based on various cell lines and animal models; further validation in clinical settings is needed. There are limited long-term effects and safety profile data.	CB shows potential as a selective and effective treatment for lung cancer, with promising in vivo antitumor activity. Further research is required to confirm clinical efficacy and safety.	[50]
NSCLC cell lines (A549, H1299, H460, and SK-MES-1) in vitro and NSG mice in vivo	CB: 0.6, 1.2, 2.5, 5, 10, and 20 μM for in vitro studies; 1.5, 5, and 10 mg/kg/day via intraperitoneal injection for in vivo studies	2 μM for 40–50% inhibitive efficacy on the four cancer cells	**In vitro:** CB dose-dependently reduced viability in NSCLC cell lines effectively. CB induced significant cell cycle arrest at the G0/G1 phase and increased apoptosis in a dose- and time-dependent manner. Elevated ROS levels and decreased MMP were observed. **In vivo:** Significant inhibition of tumor growth in NSG mice was observed with CB, especially at 10 mg/kg/day. No significant cytotoxicity was observed in rat splenocytes.	In vitro models may not fully reflect in vivo tumor dynamics and immune interactions. The findings are based on only four NSCLC cell lines and one animal model, which may not accurately encompass all NSCLC types or predict human responses. The in vivo results are limited to a specific mouse model, and CB’s long-term effects and safety are not fully addressed.	CB demonstrates promising potential as a treatment for NSCLC by targeting specific apoptotic pathways, which could overcome drug resistance and improve therapeutic efficacy. Further clinical trials are necessary to validate these findings and assess the safety and effectiveness of CB in humans.	[51]

Abbreviations: Bax, Bcl-2-associated X Protein; Bcl-2, B-cell lymphoma 2; CB, Cinobufagin; DNA, Deoxyribonucleic Acid; EMT, Epithelial–Mesenchymal Transition; FAK, Focal Adhesion Kinase; FLOT2, Flotillin-2; FOXO1, Forkhead box O1; G9a, Euchromatic Histone Lysine Methyltransferase 2; IL-6, Interleukin 6; MMP, Mitochondrial Transmembrane Potential; NSCLC, Non-Small-cell Lung Cancer; p53, Tumor Protein p53; p-Akt, Phosphorylated Protein Kinase B; PDA, polydopamine; ROS, Reactive Oxygen Species; Src, Proto-Oncogene Tyrosine-protein Kinase Src; STAT3, Signal Transducer and Activator of Transcription 3.

**Table 2 cancers-16-03758-t002:** Genetic targets of CB against cancer cell lines.

RELA	GSK3B	CCT3	H2AX	CYP3A4	ATAD5
RPL6	BHLHE40	CDK2	HSP90AB1	DCTPP1	FLOT2
RRP9	BIRC2	CDK2	JUN	EGFR	ODC1
RUNX1	BIRC5	CDK9	JUNB	FOS	PARP1
SAT1	CCNA2	CDKN1A	KLF10	GRWD1	PCNA
TFB2M	TIMP2	TGFB1	MYC	TNF	TGFBR1
TYMS	FOXO1	STAT3	FAK	G9a	SRC

## Data Availability

No new data were created or analyzed in this study. Data sharing does not apply to this article.

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
