# Peer review of "Fantastic Frogs and Where to Use Them: Unveiling the Hidden Cinobufagin’s Promise in Combating Lung Cancer Development and Progression Through a Systematic Review of Preclinical Evidence"

_cancers, 2024, doi:10.3390/cancers16223758_

Round 1
Reviewer 1 Report
Comments and Suggestions for Authors
This is a intriguing topic about the story of natural compound Cinobufagin in the fight of lung cancers. The authors nearly search all possible literatures to list the potential mechanism of Cinobufagin and medical implications. I have some questions to discuss with the authors to hope to further help improve the readability of this manuscript.
1. The chemical structure of Cinobufagin should have been also modified to produce some active derivatives. Some representative Cinobufagin derivatives deserve to be also presented in the Figure 1. Also, the IC50 can be added under the structures.
2. If possible, IC50 in Table 1 can be also provided for all the models from the different literatures. The authors will easily grasp the current progress of the synthesis or derivatation of Cinobufagin structure.
3. As shown in Table 1, the drug administration route can be added.
4. In the aspect of MOA (Mechanism Of Action), some key proteins involved in cancer cell division, cell death and cell metabolism are activated or depressed, but seemingly not all occuring in all the models from the different literatures. Using cytospace or other online PPI (Protein-Protein Interaction) database tools, a protein interaction network can be added to illustrate the relationship as the the presumptive MOA model of Cinobufagin.
5. As stated in the “Conclusions and Future Research Directions”, the author refer to mitigating the potential toxicity of CB. Is ther the toxcity data to normal cells or body? If any, please also provide them for better foucus on how to design more safe and effective Cinobufagin delivery system.
6. In the “Conclusions and Future Research Directions”, antibody-drug conjugate (ADC) strategy can be also discussed when delivery systems are mentioned.
Author Response
RESPONSE TO REVIEWERS' COMMENTS
Manuscript number: cancers-3268078 ― Cancers (MDPI)
"Fantastic Frogs and Where to Use Them: Unveiling the Hidden Cinobufagin's Promise in Combating Lung Cancer Development and Progression – A Systematic Review of Preclinical Evidence"
The authors of this document wish to express their deepest gratitude to the Editor-in-Chief and the Reviewer for their thorough and insightful evaluation of our manuscript. Their expert feedback has been invaluable in enhancing the quality of our work. We have carefully considered and diligently implemented each suggestion provided, which has significantly improved the manuscript. In particular, we have made substantial revisions to address the points raised. These noteworthy changes are mainly marked with YELLOW-highlighted text throughout the document for ease of reference. Additionally, we have prepared a detailed and comprehensive response to each comment and suggestion. This response is organized in a "point-by-point" format below, ensuring that every concern has been thoroughly addressed and explained. We sincerely appreciate the time and effort invested by the Editor-in-Chief and the Reviewer, and we believe that their contributions have greatly strengthened the final version of our manuscript.
REVIEWER #1
General comment
This is a intriguing topic about the story of natural compound Cinobufagin in the fight of lung cancers. The authors nearly search all possible literatures to list the potential mechanism of cinobufagin and medical implications. I have some questions to discuss with the authors to hope to further help improve the readability of this manuscript.
General response
Dear Erudite Reviewer, thank you for taking the time to revise our manuscript and for giving us the chance to improve based on your precious comments and suggestions. After addressing all your comments and suggestions regarding our manuscript text, we are confident that a significantly improved manuscript version has emerged. We are excited to resubmit the modified version for your kind perusal and reevaluation. Thank you for your brilliant insights, important contributions, and feedback. You do have an eye for improvement. As a signal of our utmost respect for you, we want to provide you with a detailed and comprehensive point-by-point response to your comments below. Thank you once again for your time and patience in revising our article.
Comment 1
The chemical structure of cinobufagin should have been also modified to produce some active derivatives. Some representative Cinobufagin derivatives deserve to be also presented in the Figure 1. Also, the IC50 can be added under the structures.
Response
Dear Esteemed Reviewer, thank you for this insightful and important suggestion. We agree that this would be a significant addition to our manuscript. In order to improve the quality and presentation of our Figure 1 based on your precious contributions, we have adhered to your guidelines and presented in Figure 1 the derivatives of cinobufagin that mostly impacted lung cancer cells among the scientific literature alongside their respective IC50 values for 72 h. Undoubtedly, the addition has significantly impacted our manuscript's depth and discussion. Thank you for this suggestion. We are delighted to present an improved version based on your suggestions. Please refer to Lines 79-82 on Page 2 for the first mention of the modifications and Lines 94-95 on Page 3 for the additions to the figure's legend. Additionally, find the new and revised Figure 1 on Page 3.
Again, thank you for your time and expertise in revising our manuscript. We are thankful for the opportunity to address your precious input and communicate with such a critical reviewer. You have an eye for improvement. Thank you for the opportunity to revise our manuscript based on your valuable contributions!
Comment 2
If possible, IC50 in Table 1 can be also provided for all the models from the different literatures. The authors will easily grasp the current progress of the synthesis or derivatation of Cinobufagin structure.
Response
Dear Erudite Reviewer, thank you for this invaluable suggestion. You are correct, and we agree with your point. In order to improve our manuscript accordingly, we have adhered to your guidelines and added a new column to our table. Please find Table 1 on Pages 7-10 of the revised document. Although we cannot mark the corrections here using lines, we have highlighted the additions using YELLOW highlighted text for your perusal and ease of verification. The newly added information is in the third column of the revised table. We are confident that these additions have substantially increased the quality and transparency of our findings and improved the readability of our manuscript.
Again, thank you for your expertise and guidance throughout this editorial process and peer review stage. After addressing your precious comments, we are confident that our manuscript has been significantly improved. We are delighted to present you with this revised version. Thank you for everything!
Comment 3
As shown in Table 1, the drug administration route can be added.
Response
Dear Esteemed Reviewer, thank you for this comment. You are entirely correct that we must add the administration routes for cinobufagin in the in vivo experiments presented in our table. These additions would undoubtedly enhance our research's transparency, objectivity, and impactfulness. In order to improve our manuscript according to your guidelines, we have meticulously addressed your comments and addressed them within the second column of Table 1 on Pages 7-10. Although we cannot mark the corrections here about the lines they occupy, the additions were strictly highlighted by YELLOW highlighted text. All routes were intraperitoneal, and the administration was via injection. We are confident that adding this information enhanced our manuscript's quality and readability. We eagerly anticipate a positive response from you regarding this correction.
Again, thank you for your time and expertise in revising our manuscript. We are thankful for the opportunity to communicate with such a critical reviewer and address your precious input. Thank you for the opportunity to revise our manuscript based on your valuable contributions!
Comment 4
In the aspect of MOA (Mechanism Of Action), some key proteins involved in cancer cell division, cell death and cell metabolism are activated or depressed, but seemingly not all occuring in all the models from the different literatures. Using cytospace or other online PPI (Protein-Protein Interaction) database tools, a protein interaction network can be added to illustrate the relationship as the the presumptive MOA model of cinobufagin.
Response
Esteemed Reviewer, thank you for this insightful suggestion. We agree with you. However, we regret that doing a diagram with protein-protein interactions is a compendium. We tried to perform these interactions using cytospace software and did not adhere to the guidelines you expected due to the compendium. In order to improve our manuscript according to your precious input, we have compiled within the new Table 2 on Page 15 a list of genes targeted by cinobufagin in combating cancer cell lines. Please refer to Lines 405-409 on Page 15 for the introductory sentences of the new table within the main manuscript's text.
Again, thank you for your impressive eye for improvement. Communicating with such a critical and esteemed reviewer is a great honor. We are proud to present a revised manuscript based on your comments and precious suggestions.
Comment 5
As stated in the "Conclusions and Future Research Directions", the author refer to mitigating the potential toxicity of CB. Is ther the toxcity data to normal cells or body? If any, please also provide them for better foucus on how to design more safe and effective Cinobufagin delivery system.
Response
Dear Erudite Reviewer, thank you for this insightful and critical suggestion. You are entirely correct that adding more information about cinobfagin's cytotoxicity would undoubtedly enhance our manuscript's quality and transparency to the readers. In order to improve our manuscript accordingly, we have meticulously adhered to your guidelines and introduced some new information in our Discussion section to provide a more in-depth exploration of cinobufagin's toxicity against normal and cancerous lineages. We provided insights into how cinobufagin exerts more toxicity to cancer cells and provided research directions based on nanotechnology to address these potential issues. Please refer to Lines 289-317 on Pages 11-12 for the corrections.
Again, thank you for your expertise and guidance throughout this editorial process and peer review stage. We are confident that a significantly improved manuscript has emerged after we addressed your corrections and valuable contributions. We eagerly anticipate submitting the modified version and hearing from you. Thank you for the opportunity to revise our manuscript based on your brilliant input!
Comment 6
In the "Conclusions and Future Research Directions", antibody-drug conjugate (ADC) strategy can be also discussed when delivery systems are mentioned.
Response
Dear Esteemed Reviewer, thank you for this suggestion. You have an eye for improvement, and we are thankful for the opportunity to revise our manuscript according to your precious contributions. We have strictly adhered to your guidelines and implemented the necessary modifications to our manuscript's main text. Please refer to Lines 368-374 on Page 14 for the additions. We discussed the antibody-drug conjugate strategy and advocated for the conjugation of monoclonal antibodies with cinobufagin for enhanced targeted therapy via this innovative system. After addressing this comment, we are confident that our manuscript is more updated and impactful.
Again, thank you for all the essential improvements you suggested and for the opportunity to revise our manuscript according to your precious contributions. We eagerly anticipate your positive response to the revised version we are submitting. Thank you for everything!
I, the corresponding author of the manuscript "Fantastic Frogs and Where to Use Them: Unveiling the Hidden Cinobufagin's Promise in Combating Lung Cancer Development and Progression – A Systematic Review of Preclinical Evidence" under the assigned ID cancers-3268078, on behalf of my coauthors, once again extend my heartfelt gratitude to the knowledgeable Editor-in-Chief and reviewers for their time and expertise in revising our manuscript. After we addressed their constructive and refined feedback and suggestions, a significantly improved manuscript version emerged. Undoubtedly, their insightful suggestions and feedback have significantly enhanced the quality of our manuscript. We respectfully are at the disposal of the Editor-in-Chief and the Reviewer to address any additional suggestions regarding our publication. If you are satisfied with our newly refined and significantly improved version, we are eager to anticipate the acceptance of our article for publication in this most important journal, Cancers. Thank you once again for your time and expertise.
On behalf of the coauthors, the corresponding author signs on a signal of the utmost respect for the Erudite Editor-in-Chief and the Esteemed Reviewers.
Lucas Fornari Laurindo
Faculdade de Medicina de Marília
E-mail: lucasffffor@gmail.com │ lfornarilaurindo@outlook.com
The corresponding author of cancers-3268078
Reviewer 2 Report
Comments and Suggestions for Authors
This is an interesting review on cinobufagin role as an anticancerous agent. The authors have very well laid out there search criteria and having a consort diagram along with other figures are well appreciated and makes this manuscript easy to read. The article is very well written with some minor spelling mistakes. Following are some comments.
1. Authors did not dissect out the half life of this components across various articles.
2. What type of mode of delivereis can be done with Cinobufagin?
3. Is there any evidence of immunogenecity along with anaphylaxis which could be a concern while transitioning from preclinical to Phase-1 Safety and Efficacy studies?
Minor:
1. The spelling should be checked carefully in keywords Cinobufagin is spelled as cinobugafin
Comments on the Quality of English LanguageI found some spelling mistakes which should be corrected. Otherwise the article is very well writtenn.
Author Response
RESPONSE TO REVIEWERS' COMMENTS
Manuscript number: cancers-3268078 ― Cancers (MDPI)
"Fantastic Frogs and Where to Use Them: Unveiling the Hidden Cinobufagin's Promise in Combating Lung Cancer Development and Progression – A Systematic Review of Preclinical Evidence"
The authors of this document wish to express their deepest gratitude to the Editor-in-Chief and the Reviewer for their thorough and insightful evaluation of our manuscript. Their expert feedback has been invaluable in enhancing the quality of our work. We have carefully considered and diligently implemented each suggestion provided, which has significantly improved the manuscript. In particular, we have made substantial revisions to address the points raised. These noteworthy changes are clearly marked with YELLOW-highlighted text throughout the document for ease of reference. Additionally, we have prepared a detailed and comprehensive response to each comment and suggestion. This response is organized in a "point-by-point" format below, ensuring that every concern has been thoroughly addressed and explained. We sincerely appreciate the time and effort invested by the Editor-in-Chief and the Reviewer, and we believe that their contributions have greatly strengthened the final version of our manuscript.
REVIEWER #2
General comment
This is an interesting review on cinobufagin role as an anticancerous agent. The authors have very well laid out there search criteria and having a consort diagram along with other figures are well appreciated and makes this manuscript easy to read. The article is very well written with some minor spelling mistakes. Following are some comments.
General response
Dear Erudite Reviewer, thank you for taking the time to revise our manuscript and for giving us the chance to improve based on your precious comments and suggestions. After addressing all your comments and suggestions regarding our manuscript text, we are confident that a significantly improved manuscript version has emerged. We are excited to resubmit the modified version for your kind perusal and reevaluation. Thank you for your brilliant insights, important contributions, and feedback. You do have an eye for improvement. As a signal of our utmost respect for you, we want to provide you with a detailed and comprehensive point-by-point response to your comments below. Thank you once again for your time and patience in revising our article.
Comment 1
Authors did not dissect out the half life of this components across various articles.
Response
Dear Erudite Reviewer, thank you for this comment and essential suggestion. We agree with your point and are eager to anticipate your positive response to our modifications. Unfortunately, the included studies did not significantly address the half-life of the components used to treat lung cancer across the diverse models. However, in order to improve our manuscript accordingly and try to address this vital concern of yours, we have adhered to your guidelines and added to Table 1 on Pages 7-10 a new column (third column) addressing the IC50 values for each of the included models treated with cinobufagin and its formulations. We are confident that these additions will significantly enhance our manuscript's in-depth understanding and readability. Additionally, we think that our manuscript is more objective than ever after these additions. We hope you are satisfied with our corrections.
Again, thank you for your expertise and guidance throughout this editorial process and peer review stage. Thank you for the opportunity to revise our manuscript based on your brilliant input! You have an eye for improvement, and we are confident that a significantly improved manuscript emerged after we addressed your corrections and valuable contributions. We eagerly anticipate submitting the modified version and hearing from you.
Comment 2
What type of mode of delivereis can be done with cinobufagin?
Response
Dear Reviewer, thank you for this important comment. We agree with your point and your contributions. In order to improve our manuscript and adhere to your guidelines, we want to highlight Lines 366-374 on Page 14, in which we discuss the potential delivery systems that could include cinobufagin for better treatment and target of cancer cells. We are confident that these additions significantly raised the readability and impact of our manuscript.
Thank you for your precious time and consideration. We are grateful for the opportunity to communicate with such a critical reviewer. Your contributions have been instrumental in reshaping our manuscript for the better. Thank you for everything!
Comment 3
Is there any evidence of immunogenecity along with anaphylaxis which could be a concern while transitioning from preclinical to Phase-1 Safety and Efficacy studies?
Response
Dear Esteemed Reviewer, you raised an important point here! Thank you for your attention to improvement and critical eyes. You are correct that adding this information to our manuscript would undoubtedly enhance its quality and impact throughout the research community. Previous studies have indicated that cinobufagin can cause anaphylactoid reactions by triggering the release of β-hexosaminidase and histamine via IgE-R. Because of this information and to improve our manuscript based on your essential and critical suggestions, we have implemented Lines 317-321 on Page 12 to elucidate the role of cinobufagin in flaring up anaphylaxis.
Again, thank you for your eye for improvement and attention to detail. Communicating with such a critical and important reviewer is an extreme honor. Thank you for the chance to improve our manuscript accordingly. After addressing your suggestions, we are confident that our manuscript is much more significant. Thank you for everything!
Comment 4
The spelling should be checked carefully in keywords cinobufagin is spelled as cinobugafin.
Response
Dear Esteemed Reviewer, you are entirely correct, and we agree with your point. Thank you for your attention and eye for improvement. We have meticulously adhered to your point and revised the spelling of the included keywords. After carefully checking the spelling and typography of each keyword, we can assure you that no errors remained. In order to demonstrate our willingness to improve our manuscript following your precious suggestions, please find the word "Cinobufagin" corrected within the keywords in Lines 56-57 on Page 2.
Again, thank you for all the essential improvements you suggested and for the opportunity to revise our manuscript according to your precious contributions. We eagerly anticipate your positive response to the revised version we are submitting. Thank you for everything!
Comment 5
I found some spelling mistakes which should be corrected. Otherwise the article is very well written.
Response
Dear Erudite Reviewer, thank you for your comment and for allowing us to revise our manuscript to address your concerns. We have diligently and promptly passed our manuscript through a thorough language revision and made necessary adjustments to correct some spelling and typographical errors after the first submission to this influential journal. In order to demonstrate our willingness to improve our manuscript, we have marked in PURPLE highlighted text all the corrections that were performed. They were words, expressions, and complete sentences that have been modified to strictly adhere to the resolution of your concerns and improve our manuscript with concentration, transparency, and importance. After addressing these corrections, we are confident that a significantly improved manuscript version has emerged.
Again, thank you for your kindness and expertise in suggesting the necessary revisions to our manuscript. We are thankful for the opportunity to implement the necessary revisions in our text and communicate with a critical reviewer like you. Thank you for everything!
I, the corresponding author of the manuscript "Fantastic Frogs and Where to Use Them: Unveiling the Hidden Cinobufagin's Promise in Combating Lung Cancer Development and Progression – A Systematic Review of Preclinical Evidence" under the assigned ID cancers-3268078, on behalf of my coauthors, once again extend my heartfelt gratitude to the knowledgeable Editor-in-Chief and reviewers for their time and expertise in revising our manuscript. After we addressed their constructive and refined feedback and suggestions, a significantly improved manuscript version emerged. Undoubtedly, their insightful suggestions and feedback have significantly enhanced the quality of our manuscript. We respectfully are at the disposal of the Editor-in-Chief and the Reviewer to address any additional suggestions regarding our publication. If you are satisfied with our newly refined and significantly improved version, we are eager to anticipate the acceptance of our article for publication in this most important journal, Cancers. Thank you once again for your time and expertise.
On behalf of the coauthors, the corresponding author signs on a signal of the utmost respect for the Erudite Editor-in-Chief and the Esteemed Reviewers.
Lucas Fornari Laurindo
Faculdade de Medicina de Marília
E-mail: lucasffffor@gmail.com │ lfornarilaurindo@outlook.com
The corresponding author of cancers-3268078
Round 2
Reviewer 1 Report
Comments and Suggestions for Authors
Most of all my concerns have been well addressed. It is noteworthy to see that some obvious errors can be further corrected.
1. In Figure 1,the mark "?" near the chemical structure can be removed.
2. The unit of Some IC50 values is missing. Please check again.
3. In Table 2, "cancer cell lines" may be more suitable. Are there all genes are shared by the same cancer lines? If not, the cancer cell types can be also added. If any, the gene ID in NCBI database is added, it will be more welcome.
4. Why some key genes such as Akt, FOXO1, STAT3 are not listed in Table 2? Please check again to keep the consistency with genes included in the context.
Author Response
Most of all my concerns have been well addressed. It is noteworthy to see that some obvious errors can be further corrected.
Response: Dear reviewer, thank you very much for your time reviewing this manuscript.
- In Figure 1,the mark "?" near the chemical structure can be removed.
Response: Erudite reviewer, we changed the figure according to your suggestion. Thank you very much.
- The unit of Some IC50 values is missing. Please check again.
Response: Dear reviewer, we included the units that were missing. Please see highlighted in yellow in the text. Thank you for this observation.
- In Table 2, "cancer cell lines" may be more suitable. Are there all genes are shared by the same cancer lines? If not, the cancer cell types can be also added. If any, the gene ID in NCBI database is added, it will be more welcome.
Response: Dear Doctor, we change according to you comment. Please see highlighted in yellow in line 401. Thank you for this comment.
- Why some key genes such as Akt, FOXO1, STAT3 are not listed in Table 2? Please check again to keep the consistency with genes included in the context.
Response: Dear reviewer, Thank you very much for this observation. We included the missing genes, which are highlighted in Table 2.